# Using the ADAPT guidance to culturally adapt a brief intervention to reduce alcohol use among injury patients in Tanzania

**Catherine A. Staton**[1,2]*, **Armand Zimmerman**[1], **Msafiri Pesambili**[3], **Ashley J. Phillips**[1,2], **Anna Tupetz**[1,2], **Joao Vitor Perez de Souza**[1,2], **Judith Boshe**[3], **Michael H. Pantalon**[4], **Monica Swahn**[5], **Blandina T. Mmbaga**[1,3,6], **Joao Ricardo Nickenig Vissoci**[1,2]

**1** Duke Global Health Institute, Duke University, Durham, North Carolina, United States of America, **2** Department of Emergency Medicine, Duke University School of Medicine, Durham, North Carolina, United States of America, **3** Kilimanjaro Christian Medical Centre, Moshi, Tanzania, **4** Department of Emergency Medicine, Yale University, New Haven, Connecticut, United States of America, **5** Wellstar College of Health and Human Services, Kennesaw State University, Kennesaw, Georgia, United States of America, **6** Kilimanjaro Clinical Research Institute, Moshi, Tanzania

* catherine.staton@duke.edu

## Abstract

Harmful alcohol use is a leading risk factor for injury-related death and disability in low- and middle-income countries (LMICs). Brief negotiational interventions (BNIs) in emergency departments (EDs) effectively reduce alcohol intake and re-injury rates. However, most BNIs are developed in high-income countries, with limited evidence of their effectiveness in LMICs. To address this gap, we culturally adapted a BNI for alcohol-related injury patients at Kilimanjaro Christian Medical Centre (KCMC), a tertiary hospital in Tanzania. Our study followed the ADAPT guidance to culturally adapt an existing high-income country BNI for use in the KCMC, a tertiary hospital in Tanzania. The adaptation included: 1) a systematic review of effective alcohol harm reduction interventions in similar settings; 2) consultations with local and international healthcare professionals experienced in counseling and substance abuse treatment; 3) group discussions to refine goals and finalize adaptations. The adapted BNI protocol and assessment scales ensured intervention fidelity. At KCMC, 30% of injury patients screened positive for alcohol use disorder (AUD), with a five-fold increased risk of injury, primarily from road traffic accidents and violence. A systematic review highlighted limited data on patient-level interventions in low-resource settings. Our adapted BNI, '*Punguza Pombe Kwa Afya Yako* (PPKAY)', based on the FRAMES model, showed high feasibility and acceptability, with 84% of interventions achieving ≥80% adherence and 98% patient satisfaction. PPKAY is the first culturally adapted alcohol BNI for injury patients in an African ED. Our study demonstrates our approach to adapting substance use interventions for use in low resource settings and shows that cultural adaptation of alcohol use interventions is feasible, beneficial and empowering for our team. Our study lays a framework and method for other low resourced settings to integrate cultural adaptation into the implementation of a BNI in low resource EDs.

**Data availability statement:** The authors do not have permission to share the data widely according to existing data transfer agreements. As such, we can make the data available upon request to a third party, Gwamaka William, gwamakawilliam14@gmail.com, with the approval of the National Institute of Medical Research in Tanzania and the Kilimanjaro Christian Medical University College Research Ethics Committee.

**Funding:** This work was supported by the National Institute of Health via the Fogarty International Center (#5K01TW010000-03 to CAS). The funding body has no responsibility in terms of the design, data collection, analysis, interpretation of data, and in writing of the present manuscript.

**Competing interests:** The authors declare that there are no competing interests.

## Introduction

Globally, harmful alcohol use causes an estimated 3 million deaths each year, resulting in an annual mortality rate that exceeds that of HIV/AIDS, malaria, and tuberculosis [1]. Similarly, injuries account for 5.8 million of the world's total annual deaths, making it a leading contributor to global mortality [2]. Among all deaths resulting from harmful alcohol use, 28.7% are caused by injuries [1].

Low- and middle-income countries (LMICs), specifically those in Africa, carry the greatest burden of both alcohol consumption and injury rates [1–3]. The global rate of injury deaths per 100,000 people is 80.1 [2]. However, this rate increases to 98.8 per 100,000 people in the World Health Organization (WHO) African region [1]. Similarly, the global rate of alcohol consumption is 32.8 grams of pure alcohol per day per drinker whereas in Africa this rate increases to 40.0 grams of pure alcohol per day per drinker [1]. Moreover, alcohol consumption is a leading risk factor for morbidity among 15–24 year-old males in sub-Saharan Africa and is only expected to increase in coming years [4,5].

Within Sub-Saharan Africa, Tanzania has high rates of alcohol-related injury. At Muhimbili National Hospital in Dar es Salaam, it is estimated that 46.9% of patients presenting with injuries to the emergency department (ED) test positive for alcohol [6]. Similarly, at Kilimanjaro Christian Medical Center (KCMC) the regional referral hospital in Moshi, Tanzania, we estimated that 30% of injury patients presenting to the ED were positive for alcohol [7]. In addition, the prevalence of alcohol use disorders (AUDs) in Tanzania is high. In a survey of 1954 men and women aged 15–24 years from Kilimanjaro and Mwanza regions, 10.5% screened positive for AUD [8]. In a similar survey of 899 men and women aged 18–59 years from Dar es Salaam, the prevalence of hazardous drinking was 5.7% [9]. The most recent WHO estimate places the prevalence of AUD in Tanzania at 6.8% [1]. Harmful alcohol use appears to be higher among injury patients presenting to EDs when compared to Tanzania's general population. Consequently, alcohol use reduction interventions targeting high risk ED injury patients may be an initial effective and efficient means of reducing harmful drinking and injury rates in the population.

A brief negotiational intervention (BNI) is a single session interview, lasting no more than 30 minutes, which attempts to reduce harmful post-hospitalization alcohol use [10]. In high- and middle-income countries, the delivery of BNIs in the emergency department or during hospitalization for an injury has been proven effective in reducing both post-hospital alcohol consumption and post-hospital alcohol-related injury [10–13]. However, the effectiveness of a BNI in reducing harmful alcohol use among injury patients in Tanzania is still currently unknown. Studies assessing the impact of BNIs on alcohol consumption in Africa have occurred predominantly in South Africa [14–19].

To optimize the effectiveness of a BNI in producing the desired outcome, the BNI must be sensitive to a patient's culture and environment. Patients in different regions of the world may experience vastly different stressors that influence harmful alcohol and which result from unique cultural norms or acculturation experiences. The cultural adaptation of BNIs to account for such stressors among subpopulations with specific demographics is shown to be effective in improving patient outcomes [20]. As such, the aim of this study was to use the ADAPT guidance to culturally adapt a BNI for delivery to injury patients presenting to Kilimanjaro Christian Medical Centre (KCMC) ED in Moshi, Tanzania. Here we describe the process of culturally adapting a BNI to reduce post-injury alcohol use for implementation in this patient population.

## Methods

### Setting

KCMC is a zonal tertiary referral hospital serving a population of 15 million people in Northern Tanzania [21]. Over 2000 injury patients present to the KCMC ED each year, or

an average of about 46 injury patients per week with 30% of the injury population ingesting alcohol in the 6 hours prior to their injury [7]. Moreover, it has been shown that greater alcohol consumption is associated with increased odds of injury among patients presenting to the KCMC ED [7].Currently, there are limited treatment options for alcohol-reducing interventions for patients and community members, and a lack of awareness of available resources. In many LMICs with limited numbers of mental health providers and lack of treatment facilities specifically for alcohol addiction, the task of providing alcohol treatment services is then often placed on the health system [22]. The KCMC ED is the anticipated setting for the administration of a BNI designed to reduce post-injury alcohol use.

## Ethics review

All data and processes included in this study were approved by the Kilimanjaro Christian Medical Center Ethics Committee, the Tanzanian National Institute of Medical Research, and the Duke University School of Medicine Institutional Review Board. Our feasibility and clinical trials were reported through clinical trials.gov at NCT02828267 and NCT04535011 respectively. All surveys and focus group discussions were conducted between August 1, 2016 and October 31, 2018.

## ADAPT guidance

To ensure rigor and reproducibility of our adaptation process, we adhered to the processes of the ADAPT guidance [23]. The ADAPT guidance is a 4-step process model with an emphasis on cultural relevance and stakeholder engagement. It facilitates the development of interventions that are not only contextually appropriate but also likely to be embraced by the community, ensuring greater sustainability and effectiveness. Fig 1 shows the steps in the ADAPT, alongside our specific activities for each step of the PPKAY adaptation process. Some of our data and methods from this multi-year process have already been published. In this work, we summarize and cite our publications where relevant, and otherwise report our methods and results for our process.

## ADAPT step 1: Assess rationale for intervention considering intervention-context fit of existing evidence based interventions

**Mixed-method evaluation of context.** For this project, we incorporated data from prior mixed-methods evaluations of alcohol use disorder prevalence, perspectives on alcohol use behavior, barriers and facilitators to initiating treatment for alcohol use disorder in our population at KCMC and in the region [7,19,24–27,28].

**Systematic review of evidence-based practices.** Given the stated goals of adapting a patient level intervention for alcohol harm reduction in a low resource setting, we planned and completed a systematic review to provide the specific alcohol harm reduction evidence-based interventions with effectiveness from low income or African settings [19]. Here, we summarize the evidence from the systematic review, but the in-depth methods and results are published elsewhere.

**Intervention model choice.** We accumulated patient, family, healthcare practitioner, and health policy maker input into alcohol related harm needs, resources, barriers, and facilitators. We integrated these inputs into an adapted alcohol health literacy challenge logic model (Fig 2) [29].

## ADAPT step 2: Plan and undertake adaptations

**Project team training process.** We assembled a team of individuals to inform culturally appropriate changes to the model BNI protocol. Our team consisted of local professionals

**Fig 1. ADAPT process model for the PPKAY.**

(KCMC physicians including a psychiatrist when available, KCMC nurses, KCMC research assistants, and KCMC substance use counselors, all with previous experience working with patients with AUDs) external professionals (clinical psychologists, epidemiologists, and medical doctors from the United States) all with extensive experience researching substance abuse. Upon assembling our team we held multiple instructional team meetings to ensure all team members understood the purpose, components, and delivery procedure of the model BNI protocol. Similarly, they were updated on the contextual results found in Step 1's mixed-method contextual evaluation.

**Areas for adaptation identified and adaptations developed by consensus.** Our whole project team (local and external), after learning about the intervention in depth, watched examples of brief interventions being performed in English and with actors from another context (US). Afterwards, our local project team met for two weeks (three hours daily) to identify important and impactful parts of the intervention, areas of the context which could be troublesome, and potential adaptations. When deciding solutions and definitive choice of adaptations, the team had to arrive at a consensus. If areas were confusing or concerning the team would: 1) observe examples of a brief intervention in order to understand how it might work better, 2) act out the intervention plan to see how they/patients might say it/respond, or 3) consult with an external project team in order to gain external feedback on the choices. Ultimately, a consensus was determined on which areas should be adapted as well as the proposed adaptations.

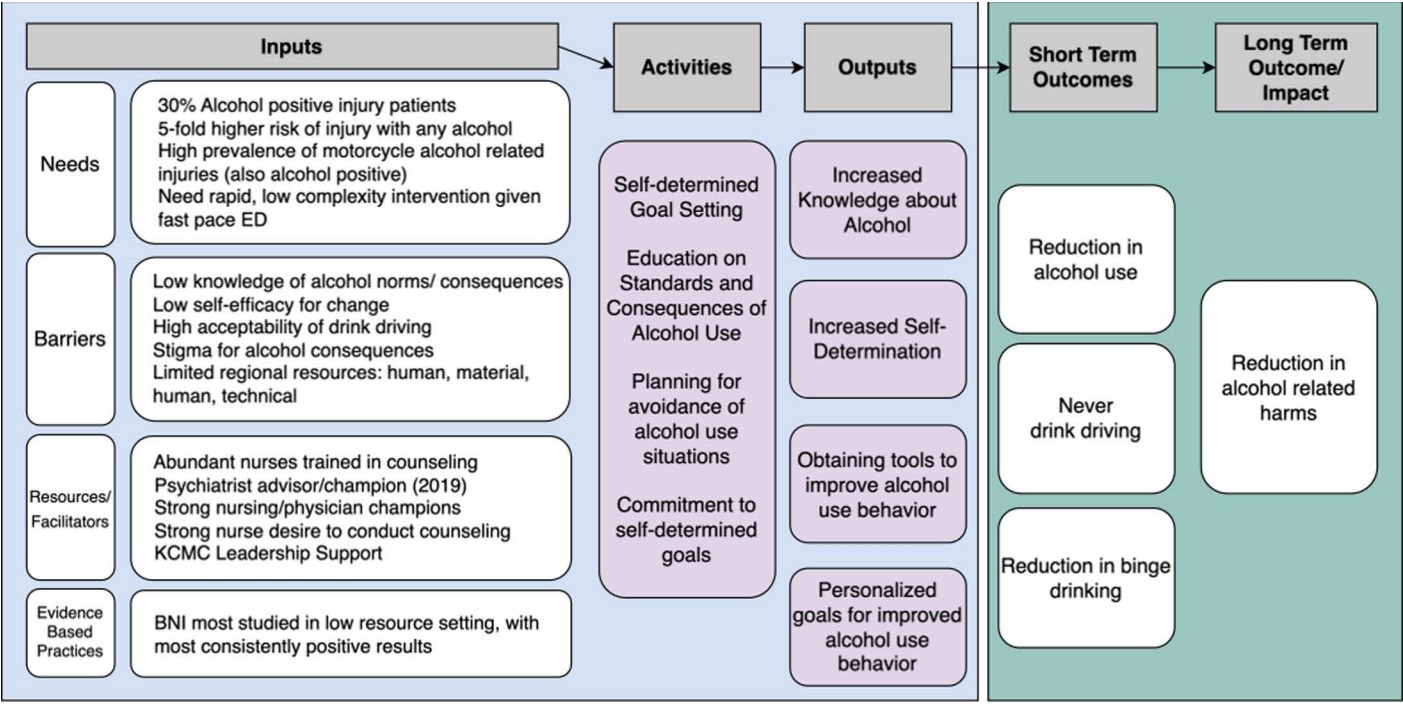

**Fig 2. Logic model of the PPKAY.**

### ADAPT step 3: Plan for an undertake piloting and evaluation

**Piloting.** We performed multiple peer assessed pilots, group-based pilots and finally patient involved pilots. First, our interventionists performed multiple interventions with colleagues who were research assistants role playing as patients they had encountered during their work with this population. Next, our interventionists performed a "good" and a "bad" PPKAY intervention in a focus group format in small groups, and with our whole research team we discussed how some parts of the intervention worked, what did not work, what could be modified and adapted. Finally, we conducted pilot interventions with patients as a final round of piloting. Piloting served two roles: first to continue to train interventionists on the processes and second to find any language, culture, and process challenges. Adaptations from piloting were incorporated with small feedback sessions where our local project ream again suggested adaptations and process changes.

**Intervention fidelity.** According to the literature, intervention fidelity has been a leading cause of intervention non-effectiveness in other pragmatic ED trials in high income settings, like the ED-SIPS trial [30]. As such, we chose to create and incorporate a rigorous fidelity plan guided by an evaluation of our adapted BNI assessment scale based on Pantalon et al.'s assessment scale [31]. Each intervention during our feasibility assessment was audiotaped and reviewed by our research members for assessment with this scale and then individual or team based feedback was given based on identified areas for improvement. We report these findings over the course of the feasibility trial.

**Feasibility evaluation.** We performed a feasibility evaluation with the goals of 1) determining the feasibility, and patient/provider acceptability of our intervention and 2) determining the feasibility of our multi-stage adaptive trial processes [24]. We conducted a 75-patient, 4 arm (usual care, PPKAY, PPKAY+ text booster, PPKAY+ personalized text

booster) pilot feasibility study [25]. These methods, including data and intervention quality metrics, are detailed elsewhere but in the currentmanuscript we will only be reporting a summary of our feasibility, and patient/provider acceptability.

**Effectiveness evaluation.** We are currently conducting an ongoing effectiveness evaluation with a pragmatic randomized clinical trial. Results are forthcoming and are not included in this manuscript [24].

## ADAPT step 4: Implement and maintain adapted intervention at scale

**Sustainability evaluation.** Understanding sustainability will be challenging without continued grant funding. We sought to pragmatically integrate the PPKAY into the care process and identify potential barriers to sustainability throughout the process. The PI (CS) also performed a focus group discussion with our Tanzanian nurse interventionists to understand possible sustainability barriers.

## Data collection and data analysis

**ADAPT step 1.** Data collection and analysis for Step 1 refers to prior published data. In summary, we conducted 1) international surveys of readiness to implement alcohol harm reduction interventions [26] 2) community advisory board, injury patient, and family member informed qualitative assessments of alcohol culture in the region [22] 3) healthcare provider perceived barriers and facilitators to conducting alcohol harm reduction interventions in the ED [27] and 4) patient and family member informed surveys and focus groups on alcohol harms and risk behavior reporting practices [7,32–36]. Our systematic review and meta-summary methods are reported elsewhere and identified the evidence based practices for similar resourced settings [19].

**ADAPT step 2.** We invited our local project team to provide feedback through a combination of personal invitation from the United States PI (CS) or the Tanzanian PI (BTM) and invitations from our paid research nurses or research assistants. Our research nurses included three female nurses with over ten years of experience including one nurse with over twenty-five years of experience who trained a significant number of nurses and doctors in the hospital. We also had one male research nurse affiliated with our team but on a different project available to do the interventions if needed. Our team composition varied based on clinical commitments but was always composed of nurses (research and clinical), physicians, advanced medical officers (mid-level providers in the Tanzanian environment), and the United States PI (CS). Open access examples of brief interventions and role-playing processes were integrated into the later training periods to serve as starting points for adaptation discussions. Focus group discussions were co-led by the United States PI in English and a bilingual research assistant; frequently discussions changed from English to Swahili for the bilingual research team to continue the conversations needed to ensure comprehension and consensus. Consensus was achieved on each step of the intervention one at a time. Notes were kept by the United States PI and research assistant.

**ADAPT step 3.**

<u>Intervention piloting:</u> Piloting was completed by interventionists with the research team, with objective observers and with external local evaluators (Tanzanian physicians and counselors) in a focus group format. Feedback was integrated by interventionalists into the adaptations in a summary meeting where notes and perspectives of the piloting were discussed and integrated into final adaptations listed in results.

<u>Intervention fidelity:</u> Each intervention per our feasibility trial (n = 60) was audio recorded and listened to by a bilingual research assistant to evaluate with our adapted BNI assessment scale. These results are listed in means with summary, standard deviations, and ranges for each scale response and analyzed over time during our trial.

Feasibility evaluation: Our feasibility evaluation is reported in full elsewhere but in brief, acceptability of the intervention was assessed from feasibility study participants in the intervention arms with a 15-question survey/interview at six months. We asked five questions about satisfaction with structured response options and four open-ended questions on the overall structure and content of the intervention [25].

During post-trial debriefing, we conducted a focus group discussion with our research nurses and research team about the complexity of the intervention, eagerness to administer PPKAY, and workload. This focus group was audio recorded and transcribed. We used inductive coding to understand the nurses' perceptions of patient acceptability, provider acceptability and perceived impact [25]. We are currently in our effectiveness evaluation with a pragmatic adaptive clinical trial and these results will be reported separately [24].

**ADAPT step 4.** Quantitative data were collected through the 75-person feasibility trial to understand potential barriers to care including enrollment, reasons for non-enrollment, who and how the intervention was performed, timing and quality of interventions and deviation over time, referrals to inpatient and outpatient psychiatric services, and attendance at that clinic. Qualitative data were collected through a focus group discussion of research team members in Tanzania to discuss barriers and solutions to sustainability of a quality intervention. The focus group discussion was performed in English or Swahili as requested by participants, transcribed and translated into English, and we conducted a thematic analysis to reach our ultimate themes and sustainability solutions.

## Results

### ADAPT step 1

**Mixed-method context evaluation.** Overall, our findings mirror regional data which find a significant alcohol use disorder burden with large gaps in resources including human, knowledge, material and technical, which was prevalent in the East Africa region [26]. We found that 30% of injury patients presenting for care at KCMC screened positive for an alcohol use disorder; we found a dose dependent increased risk of injury with an overall five fold increased odds of injury [7]. We found some evidence of inaccurate self-reporting of alcohol use [7], and a possible benefit of family reported alcohol use behavior [34] but limited alcohol testing resources in the region to support screening and treatment processes. Overall, 70% of those who were alcohol positive were injured by road traffic injuries followed by violence in 21%; and of road traffic injuries, motorcycles were common at 45% [7]. We found three classes of drinkers with 25% of participants having moderate and 15% pervasive risky behavior suffering a host of alcohol related consequences [35]. We found evidence that injury and violence sufferers believed that there was a link between alcohol and violence but there was also an external locus of control in regards to alcohol behavior and consequences. Cultural attitudes towards drinking demonstrated tribe and gender-based acceptance and even promotion of drinking among Chagga men as a social currency and status symbol, but with a strong stigmatization against women drinking in public due to traditional gender roles and stigmatization against and disapproval of those who suffer alcohol related consequences including drunk driving. Similarly, there is also a prominent frustration of the perceived inability to change the occurrence or impact of drunk driving [22]. We found that while ED health care providers all had very positive attitudes towards addressing excessive alcohol use, they had very poor knowledge of recommended alcohol use norms or limits, alcohol treatment or intervention methods, and significant stigma against those with excessive alcohol use which is reduced with more advanced education on the topic [7,22,26,27,32–35].

**Systematic review.** We reviewed all evidence for patient level interventions with efficacy testing in low- and middle-income settings [19]. While different outcomes and interventions made a meta-analysis inappropriate, our meta-summary found that brief interventions based on motivational interviewing principles (WHO-based BNI guideline or an SBIRT/US model) were the most studied with the most consistently positive evidence. There were few interventions evaluated for an African setting and those that were, were in the higher resource South African setting. Similarly, there were few ED focused interventions reported.

**Intervention model choice.** Based on these initial data, we choose the National Institute on Alcohol Abuse and Alcoholism (NIAAA) Screening and Brief Intervention and Referral to Treatment (SBIRT) for a United States ED setting. This brief intervention incorporates the Feedback, Responsibility, Advice, Menus of behavior change, Empathy, and Self-efficacy (FRAMES) model of motivational interviewing developed by Miller and Sanchez [37] into a four-step protocol [38]. We used this BNI as a starting point for creating a culturally adapted BNI for injury patients in Tanzania. Similarly, we used Pantalon et al.'s scale to measure BNI intervention adherence and modified it for our intervention and setting [31].

## ADAPT step 2

Our team composition changed daily based on clinical commitments but was always composed of nurses (research and clinical), physicians, advanced medical officers (mid-level providers in the Tanzanian environment), and the United States PI (CS). Our senior nurse and Tanzanian PI were strong champions who were very adept at enrolling members in our project team and creating a supportive implementation environment.

**Culturally adapted BNI protocol.** Using the information gained from our focus groups, we developed the 'Punguza Pombe Kwa Afya Yako (PPKAY)'/ 'Reduce Alcohol for your Health'; a one-time, 15-minute, nurse-led motivational interview discussing safe drinking behaviors and which negotiates change in alcohol use for injury patients admitted to the KCMC ED in Moshi, Tanzania (Table 1).

## Raise the subject of alcohol

**Establishing rapport.** In our focus groups, local professionals agreed that the initial greeting between patient and provider sets the tone for the entire encounter and should therefore be conducted in a way that maximizes comfort on the patient's behalf. All team members agreed that sitting if possible, introducing oneself, and inquiring if the patient's health concerns have been addressed including any pain, discomfort or hunger should be the first priority in this initial interaction. If initial interactions identified any patient discomfort or concerns, the BNI process should be postponed until the patient is more able to interact comfortably. Similarly, our context has important and at times prolonged customary greetings; as such, our first step in the PPKAY, therefore begins with these customary greetings.

**Introducing the topic of alcohol.** In our focus groups with healthcare professionals, it was mentioned that it was common for healthcare practitioners to ask about alcohol, but was, at times, stigmatizing. As such, before directing the conversation towards alcohol the provider should ask the patient for permission to discuss alcohol and we integrated the alcohol use questions in a framework of overall healthy behavior.

**Interventionist characteristics.** Local professionals in our focus groups emphasized the importance of the characteristics of the person who conducts the BNI for our target population (men and women 18 years or older). According to our focus groups, participants thought it could be inappropriate and perhaps even counterproductive to have a young male administer

**Table 1. Structured overview of the PPKAY protocol in english: step-by-step sbirt guidelines for addressing alcohol use.**

| SBIRT steps | |
|---|---|
| **1. Raise the Subject of Alcohol** | ➤ **Rapport** <br>• Sit down, introduce yourself with customary greetings <br>• Make sure patient's health-related questions/problems are addressed <br>➤ **Introduce** alcohol as the topic to discuss, ask permission to discuss |
| **2. Provide Feedback** | ➤ **Review alcohol use**, patterns and high risk patterns <br>• AUDIT testing norms for Tanzania, increased consequence/risk and personal score <br>• **R**ange of 0–40 <br>• **A**necdote: "This test has been given to millions of people worldwide, and Tanzania" <br>• **N**ormal is 0–3, 4–7 some risk, 8 + risk for injury <br>• **G**ive that person's number: XX <br>• **E**licit feedback. "What do you think about that?" <br>➤ **Identify complications (i.e., ED visit) and make a link to alcohol** <br>• Did you injure yourself due to alcohol? <br>• Have you suffered **other consequences** because of your drinking? (If none identified, bring up common complications: difficulty sleeping, challenges with finances, family problems) <br>• Help the patient **understand the link between complication and alcohol use** |
| **3. Enhance Motivation** | ➤ **Assess readiness** by the use of a ruler or 0–100% scale <br>• What is your readiness to change your alcohol use behavior? <br>• On a scale of 0 to 100, where 0 is not at all ready and 100 is extremely ready, how would you rate yourself? <br>• Ask the patient **why he or she chose this number and not a lower number** <br>• Use motivational interviewing concepts: express empathy, develop discrepancy, roll with resistance, support self-efficacy/confidence <br>➤ **SUMMARIZE** <br>• Summarize typical alcohol use, complications, readiness to change, reasons for change |

| **4. Negotiate and Advise** | ➤ **Negotiate goal(s)** <br>• Deliver MENUS of change options <br>• **M**anage your drinking (drink/day) <br>• **E**liminate drinking from your life <br>• **N**ever drink and drive** <br>• **U**sual drinking pattern (days/wk) <br>• **S**eek help (Support person/Clinic) <br>➤ Refer to patient for their choice of action | ➤ **Give advice** <br>• Ask permission to give advice <br>• Give supportive advice on their chosen action <br>• Support person, clinic |
|---|---|---|

| **4. Negotiate and Advise** (cont.) | **SUMMARIZE and formalize agreement** <br>• Summarize patients statements in words of change <br>• Emphasize patient's strengths <br>• Reiterate what agreement was reached <br>• Ask the patient if he or she has any final statements to add | |

the BNI to an older male; local professionals and patients recommended that experienced female nurses, typical of Tanzania's nursing population, would elicit the most productive feedback or discussions with patients, and would be perceived more positively. It was believed amongst our participants that, when speaking to elderly female nurses, patients are more comfortable discussing alcohol and listening to advice as they are talking to an elder like their mother/grandmother. In contrast, our focus groups also highlighted that the Muslim male population would not be comfortable discussing alcohol use with a female provider because of religious/cultural norms. While Tanzania does have a large Muslim population, Muslim males are a minority of our ED injury population. Ultimately, our team agreed that having senior female nurses and at least one male nurse available to deliver BNIs was the best option.

## Provide feedback

Cultural adaptation of this BNI step focused on 1) alcohol use patterns at the Tanzanian versus other levels, 2) use of the Alcohol Use Disorder Identification Test (AUDIT) as a marker of

normative and increased risk behavior in the Tanzanian setting, 3) culturally and linguistically appropriate Swahili words to describe alcohol use, alcohol amounts, and alcohol use disorders, and 4) the most common complications of alcohol use in our setting.

**Population-level alcohol use patterns.** To help patients compare their alcohol use to societal norms, our focus groups and patient feedback suggested it was important to relate a patient's alcohol use behavior to alcohol use patterns at the population level. Overall, our focus groups discussed how the population presenting to our health system is mostly a local population who are somewhat insulated and as such, international guidelines are less relatable to the typical patient than national or regional guidelines. Our team agreed that offering patients a description of alcohol norms from a regional perspective would allow patients to better contextualize their own alcohol-related behavior. However, our team suggested allowing the discussion around alcohol norms to remain malleable. The conversation should begin with regional alcohol norms but can shift to national or international norms if the interventionist feels these norms are more in line with the patient's own alcohol-related behavior. As an additional means of helping patients contextualize and quantify their alcohol use within the regional population, we created a visual guide to standard drinks (Fig 3) that can be shared during the discussion of population-level alcohol use patterns.

**Administration and explanation of the AUDIT.** Our team has previously conducted multiple validations of the AUDIT in Tanzania's injury patient population to ensure both content and clinical validation[32]. Our research team decided the AUDIT was the best way to screen ED patients for hazardous and harmful alcohol consumption due to both its quantity and consequence measurement components. Next our adapted BNI protocol offers feedback regarding their personal AUDIT score. To facilitate discussion regarding the AUDIT, our

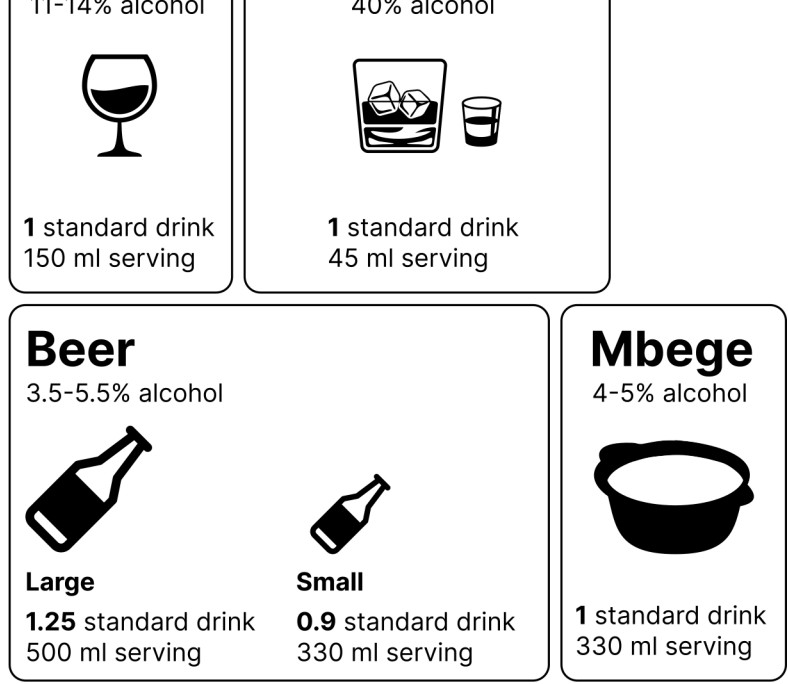

**Fig 3. Our standard drink guide for the Tanzanian setting.**

team used the Range, Anecdote, Normal, Give, Elicit (RANGE) mnemonic to assist nurses in discussing AUDIT results. In the RANGE model, the interventionist states the range of possible AUDIT scores, provides an anecdote exemplifying the AUDIT's validity, states the range of scores corresponding to normal alcohol use, reveals the patient's AUDIT score, and elicits feedback from the patient by asking what he or she thinks of their score.

**Linguistic adaptations.** In our focus groups, linguistic and culturally appropriate non-judgmental words were chosen and enforced by our bilingual team performing the intervention (Table 2). We were very careful not to include any Swahili translations of English words regarding alcohol which might carry negative connotations.

**Common complications of harmful alcohol use.** Local professionals in our focus groups verified patient feedback (ADAPT step 1 data) that finances, relationships/family, work/employment, and health are the most common areas of life in which alcohol creates complications among patients in our population. At this point, the interventionist shares these common alcohol-related consequences with the patient. The purpose of sharing these consequences is to encourage the patient to consider and discuss how alcohol adversely impacts his or her own life.

## Enhance motivation

The third step of the model BNI protocol assesses a patient's readiness to change their alcohol-related behavior and incorporates open-ended questions about their readiness to change as well as reflective listening. Cultural adaptation of step 3 focused on 1) the development of a readiness to change ruler appropriate to the setting, 2) making a link between alcohol-related consequences and a patient's alcohol use, 3) making a link between a patient's personal problems and their alcohol use, and 4) using motivational techniques to encourage behavior change.

**Developing a readiness to change ruler.** All members of our focus group decided that a 0–100 scale ranging from "not at all ready" to "extremely ready" was appropriate for

Table 2. Swahili translations and contextual considerations for English alcohol-related terms.

| English words | Swahili translations and concerns | Final Swahili translations used |
|---|---|---|
| Alcohol | Pombe - most common word for alcohol, captures all forms of alcohol.<br>Kileo - too uncommon.<br>Kilevi - too uncommon. | Pombe |
| Alcoholic | Mnywaji - term for someone who is a normal social drinker.<br>Mnywaji wa kupindukia - term for someone who is a heavy drinker.<br>Mlevi - implies someone's alcohol consumption is a problem, too labeling.<br>Chapombe - implies someone is a severe alcoholic, too labeling. | Mnywaji<br>Mnywaji wa kupindukia |
| Alcohol Use Disorder | Mraibu wa pombe - term used in the medical field to diagnose someone with an alcohol use disorder, not typically understood by the public.<br>Changamoto zinazotokana na utumiaji we pombe kupita kiasi - term for alcohol use disorder that is more generally understood by the public.<br>Matatizo ya matumizi ya pombe - term is more appropriate for expressing that someone has an alcohol related problem, but is generally understood by the public. | Mraibu wa pombe<br>Changamoto zinazotokana na utumiaji we pombe kupita kiasi<br>Matatizo ya matumizi ya pombe |
| Alcohol complications | Madhara ya utumiaji wa pombe - commonly used to express alcohol complications, has no negative connotations. | Madhara ya utumiaji wa pombe |
| Alcohol Abuse | Matumizi mabaya ya pombe - term for alcohol abuse, no negative connotations. | Matumizi mabaya ya pombe |
| Hangover | Hangover - English word is commonly used in our Swahili speaking setting. | Hangover |

assessing readiness to change in our setting. Initially, some of our external professionals opposed a 0–00 scale, and proposed a 0–10 scale, given the nature of the question. A self-evaluation of readiness to change requires significant thought and reflection. A 0–100 scale presents too large a range of possibilities to choose from which may hinder or stall the goal of assessing readiness. However, local members expressed strong support for a 0–100 scale because numerical expressions in terms of percentages are more common in Tanzanian society. Even at basic schooling levels, tests are scored on a 0–100 scale, thus being more universally known.

External professionals in our focus groups also noted that the goal of assessing readiness to change is to specifically identify why an individual patient thinks he or she needs to change. Consequently, we highlighted the following question to supplement the readiness to change ruler: **"why did you not pick a lower number?"** Such a question encourages the patient to defend their reasons for being ready to change, rather than the contrary, "why did you not pick a higher number?" which highlights barriers to change.

**Establishing a link between alcohol use, its consequences, and personal problems.**  Focus groups agreed that the link between alcohol use and alcohol-related consequences, such as injuries, is not obvious to many patients in our setting. Our team concluded there is a general lack of understanding in our setting regarding safe drinking limits; similarly, there is a large barrier to understanding the difference between causation and association of alcohol use and alcohol consequences. The third step of our PPKAY, therefore, establishes the link between harmful alcohol use and alcohol-related consequences, discusses ways that alcohol can be associated with consequences as opposed to a causative process, and discusses the patient's personal experiences with use and consequences. If a patient does not understand the connection between their alcohol use and their personal problems, then he or she is unlikely to be receptive towards behavior change.

**Encouraging behavior change.**  Focus groups revealed that care in the Tanzanian setting is often paternalistic with physicians providing specific instructions in the form of commands (i.e., "do this" or "do not do this"). Encouraging or empowering, rather than telling, patients to change their behavior is uncommon. Consequently, the third step of our PPKAY ends with the interventionist summarizing the discussion from this step using motivational techniques outlined in the model BNI protocol. These techniques include expressing empathy towards the patient, developing discrepancy between the patient's alcohol use behavior and good health, "rolling with" a patient's resistance to behavior change so as to avoid confrontation and elicit productive feedback, and supporting the patient's self-efficacy and confidence to build motivation.

## Negotiation and advice

The final step of the PPKAY creates a plan for the patient to reduce harm resulting from alcohol use, offers advice to the patient regarding their alcohol use, and formalizes an agreement between the patient and the interviewer. Cultural adaptation of step 4 focused on 1) developing an array of patient options for reducing alcohol-related harm, 2) discussing the most effective way to offer advice to a patient, 3) discussing the most practical way to formalize an agreement with a patient, and 4) deciding how to summarize the results of the intervention.

**Reducing alcohol-related harm and offering advice.**  Our focus groups agree with the BNI suggested MENUS of change options: **m**anage drinking, **e**liminate drinking from your life, **n**ever drink and drive, reduce your **u**sual drinking pattern, and **s**eek help. In our setting, given the cultural acceptance and prevalence of drink driving [22,33,35], the limited regulations our project team chose to highlight the 'never drink and drive' option for all patients in addition to their personal choice of action. It is important to note that each of these

options may not apply to each patient. Thus, options should be modified and presented to individuals at the discretion of the interventionist.

Members of our focus groups agreed asking patients for permission to offer advice is not necessary because the intervention implies that professional advice will be given. However, we included asking for permission to give advice in step four of our adapted BNI protocol as a way of keeping the patient comfortable and in control of the intervention.

**Formalizing an agreement.** To reinforce a patient's commitment to behavior change, all focus group members agreed that a verbal agreement was most appropriate as many patients in our Tanzanian setting have low education and as such relatively limited writing capacity and a low literacy rate. Focus group members also agreed that asking patients to identify a potential supporter of change should be part of the agreement process to make patients feel like they do not have to confront behavior change alone. However, identifying a personal supporter of change is entirely optional as some patients may feel uncomfortable sharing private information with family or friends. In our setting, a referral to other treatment options is an intermittently available option, becoming more available with the establishment of different mental health clinics in the region, so referral to these options has been increasing. We have assembled a locally relevant list of alcohol treatment resources and potential costs for patients for further care and support in this process.

**Summarizing results of the BNI.** The final element of the PPKAY is to present a summary of the intervention that emphasizes the patient's statements, strengths, and agreement. All members of our focus group agreed that the summary should end by giving control back to the patient. Following the summary, we, therefore, added the following question: "Are there any additional statements you would like to add?" By asking this final question, the patient has the opportunity to end the intervention with their own concluding remarks.

## ADAPT step 3

**Intervention fidelity.** In total, 65 of the 75 participants who were enrolled in our feasibility evaluation took part in the PPKAY intervention and each of these were administered by one of our three female nurse interventionists (Table 3). Overall, 84% (n = 49) of the interventions had an 80% or better adherence to the PPKAY intervention assessment scale. During the feasibility evaluation, our intervention fidelity improved dramatically at about the 10th BNI performed. The most commonly missed component on the PPKAY intervention assessment scale was the 20th item with 55 (95%) wrong answers, "Redirect statement that disfavor change." With frequent feedback to our interventionists most challenges in our intervention fidelity were resolved.

## ADAPT step 4

**Sustainability.** During this intervention and implementation planning development we identified numerous ways to support sustainability. There is some personnel/nurse turnover in the ED as well as swapping of nurses from department to department. As such, we have established nurse champions in multiple wards who are adept at performing the intervention. Similar to other interventions administered in the KCMC health system, a modest nurse reimbursement for interventions performed was suggested, at 2000 Tanzanian shillings or less than $1.00 for a quality intervention performed. Understanding that we were adding screening and referrals for patients who screened positive for an AUD, we continued to monitor both numbers of inpatient and outpatient referrals as well as those who showed up to clinics for our health system balance measures. Finally, we implemented a strong intervention fidelity system to reduce intervention drift especially given the anticipated large number of nurses who were conducting this intervention.

**Table 3. Evaluation of patient and provider acceptability and feasibility of the PPKAY intervention.**

| Patient Acceptability | Perceived effectiveness | 98% (n=52) of patients believed PPKAY had a large positive impact on their drinking | 2% of patients believed the PPKAY had a positive impact on their drinking |
|---|---|---|---|
| | Timing of PPKAY | 86% stated PPKAY took place at a convenient time | 14% said they were in too much pain to complete the PPKAY |
| | Interventionist characteristics | 100% of respondents believed nurses should administer this intervention | |
| Provider Acceptability | Attitude towards PPKAY | Nurses "really like[d]... the counseling process" | Increased confidence in PPKAY administration and assured clinical nurses "will have time" to conduct the PPKAY |
| | Perceived patient participation | Believed patients were interested in receiving knowledge and participate in process | |
| | Perceived burden | Nurses believed the PPKAY not burdensome to conduct | |

## Discussion

There are few culturally adapted alcohol harm reduction brief interventions suitable for an ED setting of a low-resource setting. Our Tanzanian ED setting has both limited resource availability as well as a high burden of alcohol use disorder and alcohol related consequences. Through a multiphase iterative process driven by local patient input, perspectives, experiences and preferences, and focus groups from local healthcare professionals and supplemented with guidance from external professionals, we culturally adapted a BNI to the Tanzanian setting, culture, and language. To our knowledge, this is the first AUD intervention to be adapted specifically for acute injury patients in an LMIC population.

### The need for cultural adaptation

The prevalence of alcohol abuse in the Kilimanjaro region of Tanzania is among the highest in the country, exceeding values representative of other regions as well as the entire nation [1,8,9,39]. The even larger prevalence of self-reported or clinician-detected alcohol use among injury patients seeking emergency care highlights the need for evidence-based AUD interventions that can be implemented in Tanzanian EDs [6,7]. To be effective, such interventions require adaptation to the local cultural context. Given the influence of one's cultural environment, demographic factors such as age and gender are very important and need to be considered due to the sensitive or stigmatizing topics to be discussed [40–43]. Our study demonstrates that cultural adaptation of a substance use intervention can encompass a broad range of changes to the original intervention protocol. These changes may determine who administers the intervention, how a patient is greeted, how the topic of substance abuse is raised, how a patient is informed of their harmful substance use, how graphics are visualized within the intervention protocol, how behavior change is motivated, and which behavior changes are encouraged. While such changes may not be required in every unique cultural setting, the results of our study provide a framework that may help construct cultural adaptation procedures for substance use interventions in other LMICs.

Cultural adaptations have been applied to BNIs for non-Tanzanian populations at increased risk of harmful substance use, and have indicated high participant satisfaction [44,45]. In addition, culturally adapted substance use interventions have proven to be more effective than their non-adapted counterpart in reducing harmful behavior in a multitude of settings including both high-income countries and LMICs [46–48]. In Tanzania, however, there are no culturally sensitive alcohol use interventions that have been systematically developed and established across healthcare facilities [22]. Our PPKAY tool is the first substance use intervention to be culturally adapted for administration to Tanzanian patients who test positive for harmful alcohol consumption. The numerous changes deemed necessary to make

the original intervention protocol applicable to Tanzanian patients highlight the need for rigorous cultural adaptation of interventions developed outside of LMICs, but applied to LMIC populations.

## The cultural adaptation process

Cultural adaptation refers to modifications of an evidence-based intervention that render the intervention more compatible with the cultural, social, and linguistic behaviors of a target population [49]. The adaptation process, therefore, relies on community engagement to support the incorporation of these societal norms into the adapted intervention. Notably, a community advisory board (CAB) is commonly used throughout the iterative adaptation process to ensure appropriate cultural translation of a substance use intervention without compromise of the intervention's core effective components [41,50–52]. In the Tanzanian setting, the use of a CAB alone to guide the adaptation of our intervention was challenging due to limited knowledge among the target population regarding safe alcohol use behaviors, AUD interventions, and local AUD treatment options [22,27]. Consequently, our approach integrated over five years of prior patient mixed-method data and a team of local healthcare professionals with extensive experience in substance use counseling supported by external professionals with extensive substance use research experience to maintain fidelity of the original intervention protocol throughout the adaptation process. This approach is not entirely unique; Ornelas et al, and Whitty et al, describe provider/patient and provider methods of intervention adaptation for sub-populations in high risk groups of high-income settings [53,54]. However, our study integrates significant amounts of mixed-method patient-focused data, the collective knowledge of community members, local healthcare professionals, and external professionals creates a rich environment for cultural adaptation of substance use interventions in LMICs, even in settings where community-based participatory research is less common. While our cultural adaptation process has many strengths, it is important to highlight a few limitations. The use of the ADAPT framework presents challenges that were encountered during our adaptation process. The framework can be resource-intensive, requiring significant time and personnel investment, which may not always be available, particularly in low-resource settings such as Tanzania. Additionally, the dependence on extensive mixed-method data and the expertise of both local and international teams, can introduce biases and constraints, particularly if the existing literature and evidence are not entirely reflective of the local context's nuances.

## Next steps

Our PPKAY intervention was designed with the entirety of our target population in mind. In its current form, our BNI seeks to reduce harmful alcohol use for any injury patient with an AUD presenting to KCMC in Moshi, Tanzania. However, the Kilimanjaro region of Tanzania is incredibly diverse with regard to age, education, income, religion, tribe, and other socio-demographic factors. Intragroup diversity of this kind may create differences in intervention efficacy between subgroups [49]. As we implement our adapted BNI in the Tanzanian setting, it is important that we compare the effectiveness of our intervention overall as well as across these subgroups so that we may continue tailoring our BNI to individuals with specific backgrounds. Furthermore, there is substantial stigma across Tanzanian communities in the Kilimanjaro region towards individuals who exhibit harmful alcohol use [22,33,36,28]. External and internalized stigma may prevent patients from engaging in conversations elicited by our BNI. It is therefore important that we ensure our intervention can create meaningful behavior change among those who feel stigmatized.

## Conclusion

We adapted a BNI developed to lower the risk of alcohol consequences among ED patients in the United States and adapted it through a multiphase iterative process to create an AUD intervention applicable to ED patients in the Tanzanian setting. Our adapted tool, the PPKAY, is a 15-minute brief intervention that uses motivational interviewing techniques to motivate behavior change among injury patients who engage in harmful alcohol use. Our study outlines the specific steps taken to make the cultural adaptations of this BNI. Our study also demonstrates an approach for how cultural adaptation of substance use interventions can be made in low-income settings as well as settings where community knowledge surrounding substance abuse is limited. Given the scarcity of research on alcohol use in Tanzania and similar settings, we encourage others to explore the cultural adaptation and evaluation of interventions to reach populations in need. Efficacy analysis of this adapted BNI is ongoing at KCMC and results will be reported in a future study.

## Supporting information

**S1 Checklist.  Inclusivity in global research.**
(DOCX)

**S1 Text.  Reporting guideline.**
(DOCX)

## Author contributions

**Conceptualization:** Catherine A. Staton, Ashley Phillips, Anna Tupetz, Monica Swahn, Blandina T Mmbaga, Joao Ricardo Nickenig Vissoci.

**Data curation:** Msafiri Pesambili, Anna Tupetz, Judith Boshe, Michael Pantalon, Blandina T Mmbaga, Joao Ricardo Nickenig Vissoci.

**Formal analysis:** Armand Zimmerman, Ashley Phillips, Anna Tupetz, Joao Ricardo Nickenig Vissoci.

**Funding acquisition:** Catherine A. Staton, Michael Pantalon, Monica Swahn, Blandina T Mmbaga, Joao Ricardo Nickenig Vissoci.

**Investigation:** Catherine A. Staton, Ashley Phillips, Anna Tupetz, Blandina T Mmbaga, Joao Ricardo Nickenig Vissoci.

**Methodology:** Catherine A. Staton, Armand Zimmerman, Anna Tupetz, Judith Boshe, Blandina T Mmbaga, Joao Ricardo Nickenig Vissoci.

**Project administration:** Armand Zimmerman, Msafiri Pesambili, Ashley Phillips.

**Resources:** Blandina T Mmbaga.

**Software:** Armand Zimmerman.

**Supervision:** Catherine A. Staton, Ashley Phillips, Anna Tupetz, Judith Boshe, Monica Swahn, Blandina T Mmbaga, Joao Ricardo Nickenig Vissoci.

**Validation:** Anna Tupetz, Joao Vitor Perez de Souza, Michael Pantalon, Monica Swahn, Blandina T Mmbaga.

**Visualization:** Armand Zimmerman.

**Writing – original draft:** Catherine A. Staton, Armand Zimmerman, Anna Tupetz, Joao Vitor Perez de Souza.

**Writing – review & editing:** Catherine A. Staton, Msafiri Pesambili, Ashley Phillips, Judith Boshe, Michael Pantalon, Monica Swahn, Blandina T Mmbaga, Joao Ricardo Nickenig Vissoci.

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
