## [Decision Letter · Decision Letter 0]

24 Sep 2024

PGPH-D-24-00483

Using the ADAPT guidance to culturally adapt a brief intervention to reduce alcohol use among injury patients in Tanzania

Dear Dr. Staton,

Thank you for submitting your manuscript to PLOS Global Public Health. We finally got the needed revieweres to proceed with the revision of your wokr. After careful consideration, we feel that it has merit but does not fully meet PLOS Global Public Health’s publication criteria as it currently stands. Therefore, we invite you to submit a revised version of the manuscript that addresses the points raised during the review process. Please consider carefully all the information from Reviewer 1

We look forward to receiving your revised manuscript.

Kind regards,

Jose Ignacio Nazif-Munoz, Ph.D.

Academic Editor

Journal Requirements:

2. Please provide separate figure files in .tif or .eps format.

Reviewers' comments:

Reviewer's Responses to Questions

**Comments to the Author**

1. Does this manuscript meet PLOS Global Public Health’s publication criteria ? Is the manuscript technically sound, and do the data support the conclusions? The manuscript must describe methodologically and ethically rigorous research with conclusions that are appropriately drawn based on the data presented.

Reviewer #1: Partly

Reviewer #2: Yes

2. Has the statistical analysis been performed appropriately and rigorously?

Reviewer #1: Yes

Reviewer #2: N/A

3. Have the authors made all data underlying the findings in their manuscript fully available (please refer to the Data Availability Statement at the start of the manuscript PDF file)?

Reviewer #1: Yes

Reviewer #2: Yes

4. Is the manuscript presented in an intelligible fashion and written in standard English?

Reviewer #1: Yes

Reviewer #2: Yes

5. Review Comments to the Author

Reviewer #1: Review Reports

Title: Using the ADAPT guidance to culturally adapt a brief intervention to reduce alcohol use among injury patients in Tanzania

Review Comments

A. Is that adaption or adoption?If so what are adopted?What are not?What are adapted?what are not?

B.Is that tool adaption or intention to use the guideline, make sure to state it.

C.The abstract should be rewritten e.g. The methods iss incomplete and the result contains methods.

D.On the background

-It's well written

-What intervention was given before the ADAPT guideline?What is the limitations of the guideline? Why the ADAPT guideline is selected? What is is the importance and uniqueness of it?

E. On the Mrtjods section

-Lacks logical flow

-Mis use of tenses ' we will'

-Unnecessary underlining

-Niot used references for '...for prior mixed evaluation...'

-The validity and reliability is not presented

-The trustworthiness for qualitative and data quality for quantitative data was not presented.

-Not reported how the triangulation was done.

F. On the Result and the consequent section

-The systematic review is missed and the meta analysis is not presented.

-Tables are not self explanatory

-Should be clearly and briefly presented

-The discussion lacks strength for theoretically wel as practilical implications.

-Lacks some components E.g. funding, conflict of interests

Reviewer #2: The article is written clearly and objectively.

The introduction and methods are well-defined, however, the authors could improve the clarity of the article's objective. It seems to be confused with the objectives of the research as a whole.

The best-described objective would be: "to describe the process of culturally adapting a BNI to reduce post-injury alcohol use for implementation in this patient population."

6. PLOS authors have the option to publish the peer review history of their article (what does this mean? ). If published, this will include your full peer review and any attached files.

**Do you want your identity to be public for this peer review?** **********

---

## [Editor Report · Decision Letter 1]

3 Jan 2025

Using the ADAPT guidance to culturally adapt a brief intervention to reduce alcohol use among injury patients in Tanzania

PGPH-D-24-00483R1

Dear Prof Staton,

We are pleased to inform you that your manuscript 'Using the ADAPT guidance to culturally adapt a brief intervention to reduce alcohol use among injury patients in Tanzania' has been provisionally accepted for publication in PLOS Global Public Health.

Best regards,

Jose Ignacio Nazif-Munoz, Ph.D.

Academic Editor

Reviewer Comments (if any, and for reference):

Thank you for providing the requested questionnaire for global inclusivity. However, we note that the questionnaire has been submitted as an "other" file type. Kindly resubmit the global inclusivity questionnaire as "Supporting file."